# R-Loop-Associated Genomic Instability and Implication of WRN and WRNIP1

**DOI:** 10.3390/ijms23031547

**Published:** 2022-01-28

**Authors:** Veronica Marabitti, Pasquale Valenzisi, Giorgia Lillo, Eva Malacaria, Valentina Palermo, Pietro Pichierri, Annapaola Franchitto

**Affiliations:** Department of Environment and Health, Section of Mechanisms Biomarkers and Models, Istituto Superiore di Sanità, Viale Regina Elena 299, 00161 Rome, Italy; veronica.marabitti@opbg.net (V.M.); pasquale.valenzisi@iss.it (P.V.); Giorgia.Lillo@artiospharma.com (G.L.); eva.malacaria@iss.it (E.M.); valentina.palermo@iss.it (V.P.); pietro.pichierri@iss.it (P.P.)

**Keywords:** genomic instability, replication stress, DNA repair, RecQ helicases, R-loops

## Abstract

Maintenance of genome stability is crucial for cell survival and relies on accurate DNA replication. However, replication fork progression is under constant attack from different exogenous and endogenous factors that can give rise to replication stress, a source of genomic instability and a notable hallmark of pre-cancerous and cancerous cells. Notably, one of the major natural threats for DNA replication is transcription. Encounters or conflicts between replication and transcription are unavoidable, as they compete for the same DNA template, so that collisions occur quite frequently. The main harmful transcription-associated structures are R-loops. These are DNA structures consisting of a DNA–RNA hybrid and a displaced single-stranded DNA, which play important physiological roles. However, if their homeostasis is altered, they become a potent source of replication stress and genome instability giving rise to several human diseases, including cancer. To combat the deleterious consequences of pathological R-loop persistence, cells have evolved multiple mechanisms, and an ever growing number of replication fork protection factors have been implicated in preventing/removing these harmful structures; however, many others are perhaps still unknown. In this review, we report the current knowledge on how aberrant R-loops affect genome integrity and how they are handled, and we discuss our recent findings on the role played by two fork protection factors, the Werner syndrome protein (WRN) and the Werner helicase-interacting protein 1 (WRNIP1) in response to R-loop-induced genome instability.

## 1. Introduction

DNA replication is a fundamental process of the cell, and it is not surprising that defects in its execution or regulation can give rise to severe human diseases. Although, in eukaryotic cells, DNA replication is finely regulated to guarantee an accurate DNA duplication before entering mitosis [1]; however, it is constantly under attack of endogenous and exogenous factors that can interfere with fork progression. Notably, stalled forks can fold into harmful and unstable structures that are prone to collapse or break.

One of the major impediments to the moving fork is transcription [2]. The main transcription-associated structures detrimental to fork progression are R-loops [3,4], transient and reversible structures with physiological functions [2]. However, if their turnover is deregulated, they can cause a co-directional or head-on clash between the replisome and the RNA polymerase, leading to R-loop-driven replication stress [5,6].

To remove R-loops, cells have evolved a sophisticated network of mechanisms, collectively termed the DNA-damage response (DDR). Deficiency in members of these processes is often associated with a wide range of human diseases, including neurological disorders and cancer [6,7,8]. In this review, we will report an involvement of the Werner syndrome protein (WRN) and the Werner syndrome helicase-interacting protein 1 (WRNIP1), two fork protection factors, in response to R-loop-driven genome instability, providing new clues to understanding the way replication–transcription conflicts could be handled.

## 2. Replication Stress

Any event perturbing DNA replication dynamics is a potential source of “replication stress”. A common feature of this *status* is the slowdown of DNA synthesis and/or replication fork stalling, which are considered to be the cause and the consequence of replication stress. Notably, replication stress has been considered as the main source of genome instability during malignant transformation [9].

Therefore, replication stress depends on a wide variety of events including: (i) deregulated firing of replication origins; (ii) obstacles to replication fork progression; (iii) conflicts between replication and transcription machineries; and (iv) DNA replication under inappropriate metabolic conditions (unbalanced DNA replication). The over-expression or constitutive activation of oncogenes (such as HRAS and c-Myc e Cyclin E) is recently emerging as a novel replication stress source [10,11]. Incomplete DNA replication, unresolved DNA repair intermediates and intertwined sister chromatids can therefore lead to chromosome breakage due to chromosome non-disjunction [11]. 

Currently, there is evidence demonstrating a causative link between replication stress and the so-called chromosomal instability (CIN) (a typical feature of many types of cancer identified by the presence of gross chromosomal alterations) and a high rate of loss of heterozygosity (LOH), all events that represent “fuel” for tumorigenesis [12,13]. The characterisation of mutations responsible for genetic syndromes revealed the key role of proteins involved in the cellular response to replication stress. Among these are the DNA RecQ helicases WRN, BLM and RECQL4. All these proteins play an important role in the resolution of replication/repair intermediates, recovery of stalled/collapsed replication forks, and, when mutated, are responsible for human genetic diseases characterised by premature aging and cancer predisposition—all phenotypes that are strictly related to genomic instability [14]. 

## 3. Transcription as a Source of Replication Stress 

Transcription–replication conflicts (TRCs) represent a potential source of endogenous genomic instability in eukaryotic cells. Indeed, as replication and transcription take place on the same DNA template, they can collide with each other, leading to replication stress. Interference of the replication and transcription machineries can result in the formation of DNA breaks and chromosome rearrangements (Figure 1). To prevent replication forks from colliding with RNA polymerases, in higher eukaryotes, replication and transcription are tightly coordinated, and, unlike in bacteria, they occur within spatially and temporally separated domains [15]. 

Nevertheless, there are conditions and specific genomic loci at which collisions between the two processes inevitably occur. Depending on the orientation of a particular gene relative to the origin of replication, collisions between transcription and replication machineries can exist in a head-on orientation (genes transcribed from the lagging strand template) or a co-directional orientation (genes transcribed from the leading strand template) [3]. The head-on conflicts are more dangerous for fork progression because, if not rapidly resolved, they inevitably lead to fork collapse and the formation of double-strand breaks (DSBs). 

In addition to direct conflicts between transcription and replication, positive supercoils accumulate ahead of both machineries, posing an additional problem for head-on collisions. Therefore, the so-called replication fork barriers have evolved to prevent head-on collisions as it happens at the highly transcribed rDNA locus [16]. The likelihood of co-directional collision is instead influenced by the speed of RNA polymerase II (RNA pol II) versus DNA polymerase. 

On average, transcription appears to be faster than replication. Specifically, replication in human cells has an average speed of 1.5–2 kb per minute [17], whereas the transcription rate of RNA pol II in mammalian cells is around 3.8 kb per minute [18]. With an average human gene size of 10–15 kb, most genes would finish transcription within 5–10 min. By contrast, for the genes larger than 150 kb, transcriptional elongation to the end of the gene would take more than one entire cell cycle to be completed. In contrast to most other genes, long gene transcription appears to preferentially take place in G2/M, perhaps to minimize TRCs [19]. 

Moreover, human long genes harbour common fragile sites (CFS), genomic loci where DNA breakage events preferentially occur in cancer cells or in cells exposed to replication stress. Notably, these regions are hot spots for TRCs that are responsible for their intrinsic fragility [19]. *Cis*-elements that affect TRCs include changes in DNA supercoiling or DNA secondary structures, such as hairpins, triplex DNA (including hinged (H)-DNA), G-quadruplexes (G4), and R-loops.

Interestingly R-loops appear to be a major promoter of collisions between transcription and replication machineries (Figure 2). 

## 4. R-Loops as Inducers of Transcription-Associated Genome Instability 

Although R-loops have a physiological role in transcription activation and termination, as well as in replication initiation, they are a major source of transcription-associated genome instability (TAGIN) via multiple mechanisms. One reason behind this might rely on the ssDNA exposed from R-loops, which would be more susceptible to the action of nucleases and genotoxins. Nevertheless, a current hypothesis is that stabilized co-transcriptional R-loops disrupt the replication fork progression during S-phase [20]. 

This disruption can lead to fork collapse, DSBs, or incomplete replication before entry into mitosis. Indeed, the presence of R-loops during S-phase has been thought to occur due to TRCs, a phenomenon that can have severe consequences for the genome. A systematic explanation of how R-loops cause replication fork stalling is still ongoing. The roadblock hypothesis suggests that either the RNA pol II or other factors recruited to solve DNA–RNA hybrids can act as steric impediments to the moving fork. The unscheduled presence of stable R-loops causes further RNA pol II pausing on the template [21,22], ending up with TRCs. 

Very recently, it was suggested that, at least in yeast, the main obstacle to replication fork progression would be the elongating RNA Polymerase II engaged in an R-loop, rather than the resulting DNA–RNA hybrids [23]. Moreover, co-directional and head-on collisions are both capable of stabilising R-loops, although head-on collisions are believed to be more prone to cause replication fork stalling. Currently, it is thought that, in the case of head-on orientation, an R-loop might not be accessible for resolution. Conversely, co-directional movement of the two complexes would allow the fork to reach the R-loop before it is stabilised by RNA pol II pausing, thus, rendering the hybrid available for resolution or displacement by the moving replication fork. 

Another intriguing hypothesis is that R-loops can induce epigenetic changes leading to chromatin condensation [24]. Under these conditions, DNA can be more difficult to separate by the travelling fork and, thus, can lead to replication fork stalling and DNA damage, namely breaks, unscheduled recombination, and chromosomal rearrangements [7,25]. Supporting that R-loops could be one factor influencing TRCs as a source of fragility, earlier studies have reported the accumulation of R-loops at CFS [19,26]. How breaks arise following R-loop stabilisation is yet to be explained in detail. The activity of nucleases could be responsible for the generation of breaks that can promote unscheduled DNA degradation. 

A first hint towards this model was the demonstration that a deficiency in R-loop metabolism, due to the loss of Aquarious helicase (AQR) and Senataxin (SETX), activates the transcription-coupled nucleotide excision repair (TC-NER) nucleases XPF and XPG [27]. Their activity accounts for the generation of DSBs, thus, linking defective R-loop resolution with the generation of DNA damage. Deregulated R-loop metabolism was also suggested to boost oncogenic phenotype. Moreover, previous works supported the idea that stress induced by oncogenes activation could cause the accumulation of R-loops via multiple routes that, in turn, increase DNA damage. 

## 5. Mechanisms of Resolution and Removal of R-Loops 

To limit harmful consequences due to the persistence of unscheduled R-loops, cells have developed multiple systems. The mechanisms to prevent the deleterious effects of aberrant R-loops can be divided in two main processes: (i) enzymatic resolution and (ii) DNA replication/repair-coupled removal. For the resolution mechanism, the most relevant and well-known factor with this function is RNAse H, a conserved ribonuclease from bacteria to humans that specifically degrades the RNA moiety of the DNA–RNA hybrids [28]. This is a highly efficient back-up mechanism to resolve R-loops, which is supported by the observations that bacteria, yeast, and human cells depleted of or inactivated for RNAse H1 accumulate R-loops. 

In recent years, several studies have identified an increasing number of RNA-dependent ATPases that, in some cases, have been shown to possess in vitro DNA–RNA unwinding activity and, when depleted, lead to DNA–RNA hybrid accumulation. These include human SETX and FANCM and their yeast homologs Sen1 and Mph1, AQR, DDX19, yeast Pif1, DDX23, DDX1, Dbp2 (human DDX5), Sgs1 (human BLM) and others [29,30,31,32,33,34,35,36,37,38,39,40,41]. The role of RNA helicases in resolving R-loops is conceptually appealing because, in contrast to RNase H, helicases would resolve hybrids without the costly action of degrading the nascent RNA. 

In addition to cellular functions directly involved in R-loop resolution, specific DNA repair pathways might constitute a back-up mechanism to dissolve R-loops accumulated in S/G2 at stalled replication forks. In line with this, several factors deputed to the protection of stalled replication forks were also found to be important to prevent R-loop-mediated genome instability. Notably, cells depleted of Fanconi Anaemia factors, such as FANCD2, FANCA, FANCM and BRCA2 among others, accumulate R-loops and R-loop-dependent DNA breaks [41,42,43,44,45,46,47]. 

It was proposed that FANCM, which belongs to the DEAD family of DNA–RNA helicases, could actively remove R-loops [41]. However, it has also been recently shown that FANCD2 interacts with RNA processing factors, including DEAD-box RNA helicase DDX47, which could facilitate R-loop removal [47]. Both products of breast cancer susceptibility genes, BRCA1 and BRCA2, were involved in the removal of R-loops by acting directly on R-loop that can partially resemble a blocked fork [43]. 

Furthermore, the apical activators of the DDR, the ataxia-telangiectasia-mutated (ATM) and Rad3-related (ATR) kinases have been implicated in protecting the genome by sensing aberrant R-loop formation [48,49]. The list of factors linked to R-loop prevention/removal function is progressively growing and will be strongly expanded in the future. More recently, an involvement of the replication fork protection factors WRN and WRNIP1 in limiting R-loop-associated replication stress has been demonstrated in human cells [50,51]. 

## 6. A Role of WRN in Limiting R-Loop-Associated Genomic Instability 

WRN is a member of the RecQ family of DNA helicases—the only one to also possess an exonuclease activity [52,53,54]. Mutations in the *WRN* gene give rise to the Werner syndrome (WS) [55]. WS is a rare genetic disorder characterized by premature aging with an early onset of age-related diseases that is associated with chromosomal instability conferring an increased risk of cancer predisposition in humans. The *WRN* gene encodes for a protein of 1432 amino acid residues (~162 kDa), showing a localization into both nucleoli and sites where DNA replication (replication factories) and DNA repair (DNA damage foci) occur [56,57,58,59]. 

WRN participates in various important DNA metabolic pathways, such as recombination and telomere maintenance, and plays its major function in genome stability maintenance, participating in the repair and recovery of stalled replication forks [60,61,62]. 

ATR is the main kinase involved in the recognition and stabilization of stalled replication forks. To guarantee the proper handling of perturbed forks, ATR triggers the replication checkpoint phosphorylating several downstream proteins [63]. Mounting evidence has suggested an interplay between WRN and the ATR pathway [57,58,64,65,66,67]. Moreover, an ATR-dependent phosphorylation of WRN has been reported upon replication stress [58,65,67]. It has been also demonstrated that, to promote recovery from replication arrest, preventing DSBs formation at stalled forks, WRN is differently regulated by ATR and ATM kinases [65]. 

It is noteworthy that the WRN helicase activity is crucial to protect CFS from breakage and appears to be required to resolve DNA secondary structures and facilitate the replication process at genomic regions where fork stalling frequently occurs [64]. Notably, under conditions that induce CFS expression, such as treatment with nanomolar concentrations of aphidicolin, a specific DNA polymerase inhibitor, WRN acts as a mediator of the ATR-dependent replication checkpoint [64,67]. Hence, upon aphidicolin-induced mild replication stress (MRS), cells lacking WRN are unable to activate the replication checkpoint [67]. 

Consistently, the loss of WRN reproduces the typical phenotype of CFS instability of ATR-deficient cells both under unperturbed conditions and after aphidicolin treatment. In agreement with this, a model was proposed in which, in response to replication stress, ATR activation leads to cell cycle arrest to stabilize stalled forks and promote the proper recovery of replication [68]. A high-resolution method to study DNA replication dynamics at the single cell level (DNA molecular combing) has helped to identify an evident defect in replication fork progression in WS cells, which show a marked asymmetry of bidirectional forks [69]. 

This evidence suggests that replication forks stall at high frequency in WRN-deficient cells. It has been hypothesized that WRN can have a role during physiological DNA synthesis by avoiding the collapse of replication forks or by resolving fork-stalling intermediates and that loss of this ability can be decisive in the onset of the typical cellular phenotype of WRN-deficient cells [69]. An in vitro role of WRN was also reported in concert with DNA polymerase delta in promoting the rescue of replication forks stalled by unusual DNA secondary structures [70]. 

Taking into account the role of WRN in facilitating the fork progression of genomic regions affected by replication stress and given that defects of fork protection factors play a central role in the onset of TRCs [29,71,72], resulting in R-loop-driven genomic instability [73], the recent discovery of a function of WRN in counteracting R-loop accumulation is intriguing [50]. In WS cells, R-loop accumulation is counteracted by the XPG-dependent processing of DNA–RNA hybrids that causes transient DSB formation and, in turn, trigger ATM signalling (Figure 3). Therefore, the ATM pathway is instrumental in limiting R-loop-associated genomic instability in WS cells. 

In line with the observation that R-loop-mediated TRCs represents a considerable source of genomic instability and that defects in fork protection factors play a crucial role in the onset of these collisions, in WS cells, transcription-dependent DNA damage is increased upon MRS [50]. This is consistent with the elevated levels of R-loops exhibited by WRN-deficient cells. 

Although WRN helicase activity is critical for efficient rescue of DNA replication [74], it is not particularly active on DNA–RNA hybrids in vitro [75]. Therefore, WRN could act like some DNA helicases that promote the fork protection by regulating R-loop metabolism without requiring the helicase activity [37,76,77]. Based on these observations, and in agreement with previous demonstrations that unscheduled R-loops can cause replication fork stalling [20], it can be assumed that R-loop accumulation induces, at least partially, delayed replication in WRN-deficient cells and not vice versa. 

It is known that a proper checkpoint response is necessary for dealing with R-loops [78,79,80]. More recently, it was demonstrated that the depletion of ATR/CHK1 is correlated with harmful R-loop accumulation, which results in fork stalling [49]. WRN-deficient cells, but not cells expressing the helicase-dead WRN, show defective early activation of CHK1 after moderate replication perturbation [67]. Moreover, WRN plays a role in mediating ATR-checkpoint activation that counteracts aberrant R-loop accumulation [50]. Although a comprehensive knowledge of how WRN counteracts this dangerous event is not available, earlier studies can offer some suggestions on where to investigate. 

Several factors involved in removing directly or indirectly R-loops are regulated in a checkpoint-dependent manner, and, among others, the Ddx19 RNA helicase is specifically regulated by the ATR/CHK1 pathway [34]. Therefore, since WS cells exhibit defective activation of ATR/CHK1 signalling, it could be possible that Ddx19 is not relocalised to the nucleus and R-loops are not removed [34]. Alternatively, some fork protection factors, such as FANCM and BLM, which are regulated by ATR and are crucial to restrain aberrant R-loops [41,81,82,83], could not be activated in WRN-deficient cells because of impaired ATR signalling. Further investigations are currently ongoing to clarify the role of WRN in the R-loop metabolism. 

Interestingly, in the absence of WRN, an ATM-dependent CHK1 phosphorylation occurs after prolonged treatment [67]. Interestingly, the ATM pathway is mainly triggered in response to DSBs [84,85] but it also responds to stimuli that do not produce DNA breakage, such as aphidicolin-induced MRS [86,87] and R-loop accumulation in quiescent cells [48]. This allows to conclude that, in WS cells, there is a link between R-loops and ATM signalling. 

This hypothesis is supported by the observation that, although DSBs are not detected upon MRS, transcription inhibition or the removal of R-loops reduces ATM and CHK1 phosphorylation [50]. This is also corroborated by experiments from WRN helicase-dead cells. Indeed, these cells are not defective in CHK1 activation [67] neither accumulate R-loops [50]. Consistently, WRN helicase-dead cells are not sensitive to the inhibition of ATM or transcription and do not elicit ATM pathway upon MRS, unless CHK1 is inhibited. Further supporting this hypothesis, ATM associates with R-loops upon Aph treatment, and active transcription is required for this association [50]. 

However, it is also possible that the ATM pathway could be activated to repair DNA breaks deriving from R-loop processing. In line with this, loss of ATM activity in WS cells results in a transcription-dependent DSB accumulation, and a significant suppression of DSBs is observed when both XPG and ATM activities are abolished. The NER endonucleases XPG and XPF directly process aberrant DNA–RNA hybrids into DSBs [27]. 

After MRS, in the absence of WRN, XPG generates transient DSBs, leading to ATM-signalling activation and DNA repair. Similarly to what is observed in the rDNA, the most actively transcribed region of the human genome, in which persistent DNA breaks induce an ATM-dependent silencing of transcription resulting in DDR activation [88], in WRN-deficient cells, ATM might be activated not only to stimulate repair but also to limit transcription, thus, contributing to prevent massive DNA breakage. 

It is recognised that ATM and ATR kinases do not have redundant roles in human cells [89,90]. When WRN is lost, ATM limits the elevated genomic instability in the cells. Furthermore, ATM activity is localised to sites of Aph-induced fork stalling. Hence, given that depletion of CHK1, but not CHK2, destabilizes CFS [91], it is thought that the ATM-dependent triggering of CHK1 is instrumental for counteracting instability at CFS, which is already elevated in WRN-deficient cells [64,67]. In support of this, overexpression of a phospho-mimic form of CHK1 abolishes the need to activate an ATM signalling upon MRS, making the WRN function unnecessary in establishing the replication checkpoint. 

It is known that CFS are hotspots of TRCs [26] and the loss of WRN or its helicase activity results in enhanced CFS instability [64]. However, the removal of R-loops after Aph-induced replication stress significantly suppresses chromosomal aberrations in WS cells, but not in WRN helicase-dead cells. These observations are consistent with a checkpoint-dependent and independent role of WRN in preserving genome integrity in response to MRS [67]. Moreover, R-loop accumulation could contribute significantly to genomic instability in WS cells, as revealed by experiments performed by overexpressing an ectopic GFP-RNaseH1, a ribonuclease that specifically degrades the RNA moiety of DNA–RNA hybrids in the nucleus, or by transcription inhibition, which strongly reduced the levels of DNA damage and the frequency of chromosomal aberrations in WS cells. 

Hence, these findings corroborate a role of WRN in regulating conflicts arising between replication and transcription machineries. The loss of WRN leads to DSB-mediated ATM pathway activation to limit the accumulation of R-loop-associated genomic instability. Furthermore, WS is a cancer-prone disease [55]; thus, these observations provide valid evidence that critical levels of R-loops can contribute to the heightened cancer predisposition of patients. 

## 7. WRNIP1-Mediated Response Is Involved in Counteracting R-Loop-Associated Genomic Instability

WRNIP1 is a member of the evolutionarily conserved AAA+ ATPase family, originally identified as a new interactor of the well-known WRN protein by the yeast two-hybrid assay [92]. In addition to its ATPase activity, which is involved in replication fork restart, WRNIP1 also contains a ubiquitin-binding zinc finger (UBZ) domain that, even if implicated in fork-related functions, has a poorly defined function [93]. Although WRNIP1 was first identified several years ago, the precise functions of WRNIP1 are waiting to be identified. 

However, some information has come from research on yeast. The *Saccharomyces cerevisiae* homolog of WRNIP1, Mgs1 (maintenance of genome stability 1), is crucial for maintaining genome integrity [94,95]. Genetic analyses using *MGS1* mutants revealed that Mgs1 is required to prevent genomic instability caused by replication arrest [95]. Mutations of *MGS1* can enhance the accelerated aging process in budding yeast [95,96]. Over-production of Mgs1 is lethal or very toxic in combination with mutations in genes that encode proteins involved in DNA replication, such as DNA Polymerase delta (Pol δ), RFC, PCNA and RPA [94]. 

Mgs1 physically and functionally interacts in vivo with budding yeast Pol 31, the second subunit of Pol δ [97]. It was demonstrated that human WRNIP1 forms a homo-oligomeric complex that physically interacts with human Pol δ and stimulates its DNA synthesis activity, mainly by increasing the frequency of initiation events [98]. However, the homology of WRNIP1 to members of the RFC family and its ability to stimulate the DNA synthesis activity of Pol δ suggest a possible role for WRNIP1 in DNA replication and/or replication-related DNA transactions. 

Interestingly, prior studies established that WRNIP1 binds to forked DNA resembling stalled forks [99]. Subsequently, data regarding the roles of WRNIP1 in DNA transactions have been provided. These include a novel role for human WRNIP1 in stabilizing stalled forks [100]. WRNIP1 is recruited to hydroxyurea-induced stalled forks and interacts with RAD51. Indeed, WRNIP1 is directly involved in preventing uncontrolled MRE11-mediated degradation of stalled forks by promoting RAD51 stabilization on single-stranded DNA (ssDNA). Furthermore, WRNIP1 was implicated in non-canonical activation of the ATM-dependent checkpoint in response to MRS [87]. 

More recently, an interesting function of the WRNIP1-mediated response in limiting R-loop-associated genomic instability in WS cells and, more generally, in cells with a compromised ATR-mediated checkpoint response upon MRS, has been proposed [51]. ATM hyper-activation and a stable retention in chromatin of WRNIP1 are strictly associated with compromised CHK1 phosphorylation upon aphidicolin treatment, and this is consistent with the loss of ATR checkpoint activation in WS cells [50,67]. Furthermore, the ATR–CHK1 pathway has been previously involved in safeguarding genome integrity against pathological R-loops [101,102]. 

In line with this, in WS cells, the inability to activate CHK1 early after aphidicolin corresponds to increased harmful R-loop accumulation [50]. This agrees with the observation that pathological R-loops hinder replication fork movement [20,103,104] and that the depletion of ATR or CHK1 causes R-loop-dependent fork stalling [101]. WRNIP1 is required for ATM activation in response to stimuli that do not produce DNA breakage [62,86,87] and a DSB-independent but R-loop-dependent ATM pathway was reported in quiescent cells [48]. WS cells trigger R-loop-dependent ATM signalling specifically after MRS [50]. 

Consistently, an R-loop-dependent hyper-phosphorylation of ATM is detected when the ATR–CHK1 signalling is dysfunctional and chromatin-bound WRNIP1 is related to the late ATM-dependent CHK1 phosphorylation. Interestingly, compensating for a defective ATR pathway abolishes the need to activate ATM, and the degradation of R-loops weakens the association of WRNIP1 with chromatin. Therefore, every time the replication checkpoint is compromised, WRNIP1 retention in chromatin is required for establishing an ATM-CHK1 signalling, which might be engaged in limiting transcription and/or in preventing massive R-loop-associated DNA damage accumulation. 

Accordingly, ATM inhibition or WRNIP1 abrogation in these pathological contexts is accompanied by increased levels of genomic instability. The loss of both ATM and WRNIP1 potentiates DNA damage accumulation in ATR checkpoint-defective cells, thus, suggesting that WRNIP1 could play a different role than that of a mediator of ATM signalling upon R-loop accumulation. 

Of note, the accumulation of R-loops in WS cells correlates with the recruitment of WRNIP1 in chromatin. To prevent the collapse and reversal of R-loop-induced stalled forks, avoiding R-loop extension, and promoting fork restart and R-loop dissolution, BRCA2 together with other proteins are required [43]. Indeed, it has been previously demonstrated that BRCA2 facilitates RAD51 loading on ssDNA [105,106,107] and is implicated in R-loop processing [43]. Since WRNIP1 forms a complex with BRCA2/RAD51 and stabilised RAD51 to a stalled fork [100], it is possible that, in cells with replication checkpoint defects, WRNIP1 could collaborate with BRCA2 to stabilise RAD51 on ssDNA generated near/at sites of TRCs. 

In support of this, it has been observed that WRNIP1 loading in chromatin correlates with the presence of RAD51 and studies performed using a modification of the in situ proximity ligation assay, a fluorescence-based improved method that makes possible to detect protein/DNA association, revealed that WRNIP1 stimulates the association of RAD51 with ssDNA at/near R-loops upon MRS in WS cells [51]. Moreover, a defective ATR checkpoint promotes the accumulation of transcription- and R-loop-dependent parental ssDNA in the proximity of R-loops upon MRS. Thus, all these observations strongly suggest that WRNIP1 could contribute to stabilise either RAD51 nucleofilaments assembling at the displaced DNA strand of the R-loop or at the parental DNA exposed at the fork (Figure 4). 

Therefore, WRNIP1 could play two independent functions upon TRCs: stabilising RAD51 at R-loops or R-loop-dependent stalled forks and activating ATM to repair downstream DSBs derived from the active processing of R-loops and collisions with replication forks. Collectively, these findings provide new clues regarding an essential role of WRNIP1 as a replication fork protection factor, contributing to preserving the genome integrity after R-loop-driven replication stress. Importantly, since a direct connection between R-loops and cancer has been recognised [108], WRNIP1 could be employed as a target to further sensitise cancer cells to inhibitors of ATR or CHK1 that are currently under clinical evaluation.

## 8. Conclusions

In conclusion, a comprehensive mechanistic understanding of the pathways stimulating R-loop formation or contributing to their proper resolution is crucial. Indeed, TRCs are one of the most relevant sources of genome instability in human cells, and they contribute to cancer development. Moreover, the metabolism of the R-loop is also linked to the pathogenesis of “replication-related” genetic diseases, such as Schimke Immuno-osseous dysplasia, which is caused by mutations in SMARCAL1 [109] or XPF-ERCC1-related syndrome [110], which has been linked to chronic inflammation.

From this point of view, the involvement of WRN, which is related to another genetic disease characterized by premature aging and cancer predisposition, and another fork protection factor, WRNIP1, further strengthen the link between R-loop and alternative and error-prone transactions at the fork as contributors to the molecular pathology of “replication-related” diseases and cancer.

## Figures and Tables

**Figure 1 ijms-23-01547-f001:**
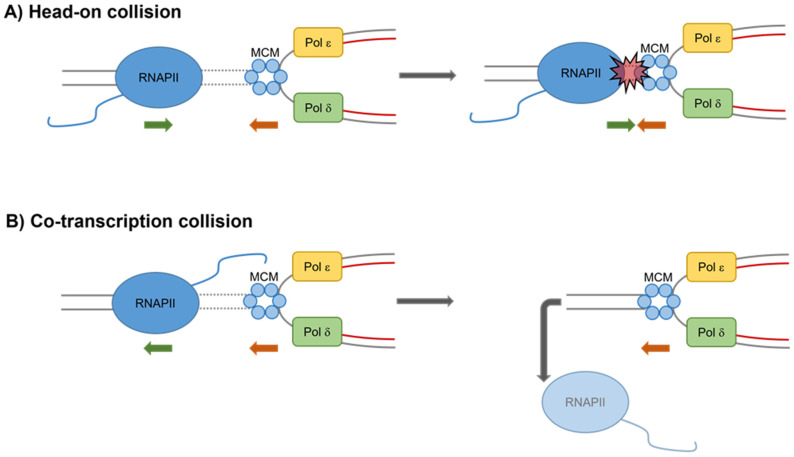
Head-on and co-directional transcription–replication collisions. (**A**) Head-on collisions are generated by progression in opposite directions of an RNA polymerase (RNA pol) and a replication fork. This leads to pausing and arrest of the replication fork that can result in fork collapse and DSBs formation. (**B**) Co-directional collisions occur when progression of an RNA pol and a replication fork are in the same direction and fork moves more quickly than the RNA pol. Displacement of the RNA pol from the DNA is required to resolve these collisions. MCM, mini chromosome maintenance complex; Pol δ, DNA polymerase δ; and Pol ε, DNA polymerase ε.

**Figure 2 ijms-23-01547-f002:**
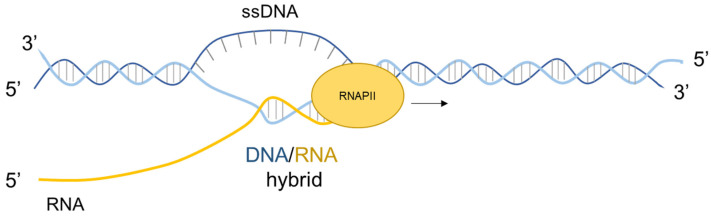
Schematic representation of the R-loop structure. An R-loop is a three-stranded nucleic acid structure composed by a DNA–RNA hybrid and a displaced non-template single-stranded DNA (ssDNA).

**Figure 3 ijms-23-01547-f003:**
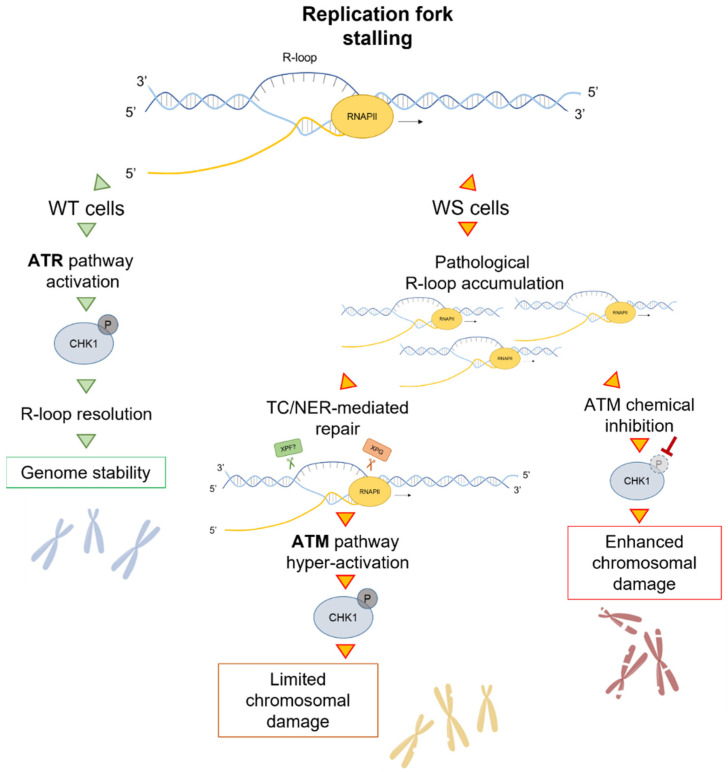
Model for a role of the ATM pathway in limiting genome instability in WS cells. After replication fork stalling, the presence of WRN leads to a proper ATR-dependent CHK1 activation that ensures R-loop resolution and thus genome stability. In WS cells, although an aberrant accumulation of R-loops is detected, a direct processing of these structures by the endonuclease XPG induces transient DSB formation, leading to ATM pathway hyper-activation. Under these conditions, ATM phosphorylates CHK1 and, thereby, counteracts chromosomal damage accumulation. Notably, if ATM kinase is chemically inhibited, the failure to respond to R-loop-associated DNA damage increases chromosomal damage, which is already elevated in WS cells.

**Figure 4 ijms-23-01547-f004:**
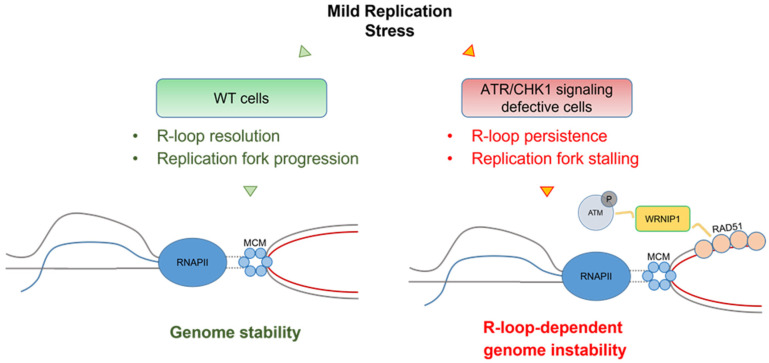
Model of the role of the WRNIP1-mediated response in counteracting R-loop-associated genome instability in cells with a dysfunctional ATR checkpoint. Following aphidicolin-induced mild replication stress in wild-type cells, proficient R-loop resolution allows proper replication fork progression, thus, ensuring genome integrity. Conversely, a defective ATR checkpoint causes R-loop persistence leading to TRCs. In this context, WRNIP1 could stabilise RAD51 at R-loops or R-loop-dependent stalled forks and contribute to the activation of the ATM pathway to repair downstream DSBs derived from the active processing of R-loops and collisions with the forks. Therefore, a dual function of WRNIP1 is required for proper maintenance of genome stability in the pathological contexts deriving from a dysfunctional ATR-dependent checkpoint.

## Data Availability

Not applicable.

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
