# Peer review of "R-Loop-Associated Genomic Instability and Implication of WRN and WRNIP1"

_ijms, 2022, doi:10.3390/ijms23031547_

Round 1
Reviewer 1 Report
This review attempts to put in context recent findings from the Franchitto lab suggesting a role for the WRN helicase and the WRNIP1 AAA+ family member in protecting against genomic instability associated with R-loop accumulation. The role of R-loops in generating DNA damage via effects on DNA replication is currently an active area of research and there are many mysteries yet to be resolved. Thus, a review on this subject could potentially be of considerable interest. I think that this review will not add a great deal to the literature on the subject, but the finding that deficiency in WRN leads to accumulation of R-loops when DNA replication is stressed is interesting and, although the significance of this observation is not yet clear, there may be some value in putting it in the context of other research on R-loops and WRN.
Here are some comments that may be useful in improving the review:
Sections 1-3 of the review deal with the replication stress and conflicts between transcription and replication. This introductory material is contained in many previous reviews and could be significantly shortened without affecting understanding of the remaining material.
Sections 4 and 5 deal with the role of R loops in promoting genomic instability and the mechanisms that remove R-loops or mitigate their deleterious effects. These sections represent the real introduction to the main discussion of the recent work of the Franchitto lab on WRN and WRNIP1. One potentially embarrassing issue is the depiction of R-loops in figures 2 and 3. In both cases the strands are mislabeled so the duplex region of the R-loop is shown as a parallel double helix (The figure shows both the RNA and DNA strands in the duplex as 5’ to 3’ in the rightward direction. The displaced DNA strand is shown as 3’ to 5’ in the rightward direction.)
Sections 6 and 7, which are the heart of the review, need more work. They are not well written and there are several statements that are hard to understand. One example is lines 309-314 – the text here is very awkward making it unclear whether the authors think that the enhanced R-loop accumulation in WRN deficiency is primarily a result of the resulting defect in checkpoint activation. Also, it was never made clear what evidence supports the view that the R-loops are causative of the genomic instability observed in these experiments, as opposed to being an epiphenomenon. Much of the discussion in these sections is highly speculative, but to their credit the authors eventually state directly that their experiments don’t yet tell us what role WRN plays in R-loop metabolism and that further work will be required (line 320). Even less is understood about how deficiency of WRNIP1 increases genomic instability – the possible role of RAD51is very speculative. Thus, the discussion is a bit unsatisfying in the end, but perhaps it can be improved with further work on the prose.
Author Response
Referee #1
Comments and Suggestions for Authors
This review attempts to put in context recent findings from the Franchitto lab suggesting a role for the WRN helicase and the WRNIP1 AAA+ family member in protecting against genomic instability associated with R-loop accumulation. The role of R-loops in generating DNA damage via effects on DNA replication is currently an active area of research and there are many mysteries yet to be resolved. Thus, a review on this subject could potentially be of considerable interest. I think that this review will not add a great deal to the literature on the subject, but the finding that deficiency in WRN leads to accumulation of R-loops when DNA replication is stressed is interesting and, although the significance of this observation is not yet clear, there may be some value in putting it in the context of other research on R-loops and WRN.
Here are some comments that may be useful in improving the review:
Sections 1-3 of the review deal with the replication stress and conflicts between transcription and replication. This introductory material is contained in many previous reviews and could be significantly shortened without affecting understanding of the remaining material.
We thank the referee for her/his useful criticisms and suggestions, and, in the revised version of our review, we have shortened the Sections 1-3.
Sections 4 and 5 deal with the role of R loops in promoting genomic instability and the mechanisms that remove R-loops or mitigate their deleterious effects. These sections represent the real introduction to the main discussion of the recent work of the Franchitto lab on WRN and WRNIP1. One potentially embarrassing issue is the depiction of R-loops in figures 2 and 3. In both cases the strands are mislabeled so the duplex region of the R-loop is shown as a parallel double helix (The figure shows both the RNA and DNA strands in the duplex as 5’ to 3’ in the rightward direction. The displaced DNA strand is shown as 3’ to 5’ in the rightward direction.)
In the revised version of our manuscript, we have amended the figure 2 and 3 introducing a new picture that depict more correctly the R-loop structure.
Sections 6 and 7, which are the heart of the review, need more work. They are not well written and there are several statements that are hard to understand. One example is lines 309-314 – the text here is very awkward making it unclear whether the authors think that the enhanced R-loop accumulation in WRN deficiency is primarily a result of the resulting defect in checkpoint activation. Also, it was never made clear what evidence supports the view that the R-loops are causative of the genomic instability observed in these experiments, as opposed to being an epiphenomenon. Much of the discussion in these sections is highly speculative, but to their credit the authors eventually state directly that their experiments don’t yet tell us what role WRN plays in R-loop metabolism and that further work will be required (line 320). Even less is understood about how deficiency of WRNIP1 increases genomic instability – the possible role of RAD51is very speculative. Thus, the discussion is a bit unsatisfying in the end, but perhaps it can be improved with further work on the prose.
In the revised version of our manuscript, we took into account the referee’s suggestion and have revised the sections 6 and 7 accordingly. We did our best to reorganize the sections with the aim to improve understanding of the text. We have also tried to better explain the remaining sections of the review.
In particular, we now have tried to put in a better context those three our key findings:
- Whenever the ATR-CHK1 signalling is dysfunctional, including in WS cells, the inability to activate CHK1 early after Aph correlates to increased R-loop accumulation (Marabitti et al., 2020).
- R-loops are causative of the genomic instability observed in WRN-deficient cells, since overexpression of an ectopic GFP-RNaseH1, which performs clearance of R-loops, decreases greatly chromosomal aberrations and DNA damage (Marabitti et al., 2019);
- Although data clarifying the precise role of RAD51 at R-loops are still missing and worth of additional investigation, RAD51 loading in chromatin correlates with the presence of WRNIP1 and, together with its association with ssDNA at/near R-loops, is suppressed by RNaseH1 overexpression (Marabitti et al., 2019). As WRNIP1 forms a complex with BRCA2/RAD51 and stabilises RAD51 to stalled forks (Leuzzi et al., 2016), it is very likely that WRNIP1 could contribute to stabilise RAD51 nucleofilaments assembling at the displaced DNA strand of the R-loop or at the parental DNA exposed at the fork.
Reviewer 2 Report
Marabitti et al have constructed a very nice, thorough literature review outlining the importance of R-loops, their occurrence and the mechanism of their resolution. Overall, I recommend accepting this manuscript after minor revision.
One minor concern with the manuscript is the fact that the title gives an impression that the review is about WRN and WRNIP and their roles in counteracting R-loops. However, the article is structured in a way that there are only a couple of sections at the end talking about these 2 proteins and the majority of review really is built up as an introduction to these.
To avoid this, I suggest either
- modifying the title by removing the emphasis on WRN and WRNIP or
- modifying the text: sections leading up to the WRN and WRNIP sections can be made shorter which may allow a stronger emphasis on the WRN and WRNIP sections. This would allow the title to fit the scope of the article.
Line 129 on Page 4: "...take more than 40 min, and thus requires more than one entire cell cycle to be completed." This sentence is bit confusing. Can the authors explain this better? if elongation takes more than 40 min, how does that equate to a timeline of more than one cell cycle?
Author Response
Referee #2
Comments and Suggestions for Authors
Marabitti et al have constructed a very nice, thorough literature review outlining the importance of R-loops, their occurrence and the mechanism of their resolution. Overall, I recommend accepting this manuscript after minor revision.
One minor concern with the manuscript is the fact that the title gives an impression that the review is about WRN and WRNIP and their roles in counteracting R-loops. However, the article is structured in a way that there are only a couple of sections at the end talking about these 2 proteins and the majority of review really is built up as an introduction to these.
To avoid this, I suggest either
- modifying the title by removing the emphasis on WRN and WRNIP or
- modifying the text: sections leading up to the WRN and WRNIP sections can be made shorter which may allow a stronger emphasis on the WRN and WRNIP sections. This would allow the title to fit the scope of the article.
We thank the referee for her/his useful criticisms and for the appreciation of our review.
We have revised the title considering the referee’s suggestion. The new title was chosen with the aim to prepare the readers for a review that reports information about R-loops and genome instability and in which the involvement of the protection factors, WRN and WRNIP1, will be also discuss.
Line 129 on Page 4: "...take more than 40 min, and thus requires more than one entire cell cycle to be completed." This sentence is bit confusing. Can the authors explain this better? if elongation takes more than 40 min, how does that equate to a timeline of more than one cell cycle?
In the revised version of our review, we have amended the text considering the issue raised by the referee.
Reviewer 3 Report
This review gives a comprehensive summary of the molecular mechanisms involved in R-loop metabolism in eukaryotes.
I have only one minor comment:
Line 44-47: R-loops can cause codirectional conflicts too. Please edit the sentence to reflect the same. Although this is mentioned later in the manuscript, I think it should be included here too.
Author Response
Referee #3
Comments and Suggestions for Authors
This review gives a comprehensive summary of the molecular mechanisms involved in R-loop metabolism in eukaryotes.
I have only one minor comment:
Line 44-47: R-loops can cause codirectional conflicts too. Please edit the sentence to reflect the same. Although this is mentioned later in the manuscript, I think it should be included here too.
We thank the referee for her/his useful criticisms and for the appreciation of our review.
In the revised version of our review, we have amended the text according to the referee’s suggestion.
Round 2
Reviewer 1 Report
The authors have taken the critique to heart and made numerous changes that improve the review, including a new and corrected figure 2. I would strongly recommend one more pass over sections 6 and 7 to improve the English. There are still a number of places where the construction of sentences is awkward. In most places the intent of the sentences is reasonably clear, but the English could be improved.
Author Response
.
